# Synthesis of Polylactic Acid Initiated through Biobased Antioxidants: Towards Intrinsically Active Food Packaging

**DOI:** 10.3390/polym12051183

**Published:** 2020-05-21

**Authors:** Marco Aldo Ortenzi, Stefano Gazzotti, Begonya Marcos, Stefano Antenucci, Stefano Camazzola, Luciano Piergiovanni, Hermes Farina, Giuseppe Di Silvestro, Luisella Verotta

**Affiliations:** 1Department of Chemistry, University of Milan, Via Golgi 19, 20133 Milan, Italy; stefano.gazzotti@unimi.it (S.G.); Stefano.antenucci@unimi.it (S.A.); stefano.camazzola@unimi.it (S.C.); hermes.farina@unimi.it (H.F.); Giuseppe.disilvestro@unimi.it (G.D.S.); 2CRC Materiali Polimerici (LaMPo), Department of Chemistry, University of Milan, Via Golgi 19, 20133 Milan, Italy; Luciano.piergiovanni@unimi.it; 3IRTA, Food Technology, Finca Camps i Amet s/n, 17121 Monells, Spain; begonya.marcos@irta.cat; 4DeFENS, Department of Food, Environmental and Nutritional Sciences-PackLab–University of Milan, Via Celoria 2, 20133 Milan, Italy; 5Department of Environmental Science and Policy, via Celoria 2, University of Milan, 20133 Milan, Italy; luisella.verotta@unimi.it

**Keywords:** polylactic acid, phenolic antioxidant, vanillyl alcohol, active food packaging, salami, shelf life

## Abstract

Polylactide (PLA)-based polymers, functionalized with biobased antioxidants, were synthesized, to develop an intrinsically active, biobased and potentially biodegradable material for food packaging applications. To achieve this result, phenolic antioxidants were exploited as initiators in the ring opening polymerization of l-lactide. The molecular weight, thermal properties and in vitro radical scavenging activity of the polymers obtained were compared with the ones of a PLA Natureworks 4043D, commonly used for flexible food packaging applications. The most promising synthesized polymer, bearing vanillyl alcohol as initiator (PLA-VA), was evaluated for active food packaging applications. Packaging with PLA-VA films reduced color and fat oxidation of salami during its shelf life.

## 1. Introduction

In the packaging field, and in food packaging above others, two tendencies have been emerging in the last few years, which have begun to spread in the consumer market. One is related to environmentally sustainable packaging, mostly relying on the use of the so-called bioplastics, as at least, partial substitutes for oil-deriving polymers. The other is related to active packaging, defined as packaging having active functions beyond the inert passive containment and protection of the product [1]. Bioplastics are generally referred to as plastics, obtained from polymers either, deriving from renewable sources or being biodegradable or both. When targeting food packaging applications, biodegradation is by far more the most important step in than the derivation of renewable sources, since the high environmental impact of food packaging plastics mainly arises from littering and incorrect end-life disposal. Research has, therefore, been pushing for the last decade towards the development of biodegradable biopolymers as substitutes for Polyethylene (PE), Polypropylene (PP) and Polyethylene Terephthalate (PET), in order to find solutions that allow biopolymers to match the properties (e.g., gas barrier or thermal stability) of traditional polymers [2,3]. Among the most promising bioplastics Polylactic Acid (PLA), that is both, biodegradable and biobased, is probably the best known and the most successful commercial example. PLA has high versatility (i.e., it can be used both for flexible and rigid food packaging) and an industrial price that is closer and closer to PE, PP or PET [4].

Also, studies and developments of active packaging have dramatically increased. The concept of active packaging can apply to a number of functions, such as antiviral [5] antimicrobial [6,7], aroma enhancer [8] or oxygen scavenging [9], the latter being one of the most exploited on an industrial level. Among the most promising kinds of active packaging techniques, antioxidant packaging, based on radical scavenging, as a mean to reduce the deterioration of packaged food, plays a major role, even if only few commercially promising examples are available [10]. Antioxidant packaging results in an increase in the shelf life of packaged food due to the scavenging of radicals produced by lipid peroxidation. A common approach relies on the introduction of the active principle as an additive in the bulk of the packaging [11,12], being either the flexible film [13] or the rigid container [14], or in a layer placed over the flexible film [15]. When the active principle is used as an additive, some drawbacks arise, the first being the migration into packaged food over time, causing possible sensory alterations [16]. A secondary drawback may lie in the potential loss of mechanical and/or optical properties of the packaging material due to the presence of the additive [17]. Another serious issue is related to the poor thermal stability of many natural radical scavengers that tend to degrade during the extrusion process, leading to undesired by-products and to the loss of their activity. Such drawbacks could be dramatically limited if the active moieties could be bonded to the polymer. Although, an extensive amount of literature is present to deal with antioxidant polymers [18,19], and to the best of our knowledge, just a few studies are present regarding the synthesis of polymers bearing the radical scavenging moieties covalently bonded to the macromolecular chains, to be used with no further treatment in the production of flexible films for packaging [20,21,22]. A similar approach has recently been used by Ge et al. to synthesize “intrinsically UV-resistant” PLAs [23], which proved very effective. In the present paper, the synthesis of intrinsically radical scavenging PLA, was investigated. Five phenolic-based antioxidants, bearing free primary alcohol moieties, and specifically Tyrosol, 3,4-dihydroxy benzyl alcohol, Pyridoxine, Vanillyl Alcohol and Ascorbic acid monoethyl ether, were added directly in the polymerization reaction feed, in order to promote the formation of a covalent bond between polymer chains and the alcoholic hydroxyl groups, while preserving the phenolic -OH and therefore its antioxidant activity. Intrinsically radical scavenging PLAs were synthesized by traditional bulk polymerization of lactide, allowing an industrially viable and inexpensive synthesis that could lead to materials having thermal properties very close to standard PLA. These features make a possible future industrial scale up extremely simple and cheap, allowing the use of standard PLA processing techniques. The phenolic antioxidants were chosen both, due to their biobased nature, easy availability, and their potential reactivity with lactide and stability in reaction conditions. The validation of the antioxidant potential of the developed material for food applications was first performed in vitro by DPPH tests; in vivo tests were later carried out on the best performing sample among the synthesized ones.

## 2. Materials and Methods

### 2.1. Materials

l-lactide, Puralact L (polymer grade), purchased from Purac Biomaterials (Gorinchem, The Netherlands) was used for the synthesis of PLAs. An industrial film-grade PLA, trademark Natureworks Ingeo^®^ 4043D (d-isomer content = 4.3% as declared by the producer, in the paper, named as “4043D”) was purchased from Resinex Srl, Italy and used as “control” PLA. Methylene chloride (DCM), methanol (HPLC purity), tin octanoate (SnOct_2_), 2-(4-Hydroxyphenyl)ethanol, known as Tyrosol (Tyr, purity > 98%), 3,4-dihydroxy Benzyl Alcohol (DBA, purity 99%) and Pyridoxine (Pyr, purity ≥ 98%), Hydrochloric acid (HCl, fuming 37%), Propyl gallate (purity ≥ 98%), Ethylenediaminetetraacetic acid (EDTA, purity ≥ 98%), 2-Thiobarbituric acid (purity ≥ 98%), Brij-35, 1,1,3,3-Tetraethoxypropane (purity ≥ 96%) were purchased from Sigma-Aldrich Co (Milan, Italy). 4-(hydroxymethyl)-2-methoxyphenol, known as Vanillyl Alcohol (VA, purity > 98%) and Ascorbic acid monoethyl ether (AA, purity > 98%) were purchased from TCI Europe N.V. (Milan, Italy). All reagents were used without further purification, and no drying process was performed.

### 2.2. Synthesis of Phenol-functionalised PLA Samples

PLA samples were synthesized in bulk using a 250 mL three-neck glass flask: 50 g of l-lactide (0.347 mol) were added in the feed together with tin octanoate (0.3% w/w on lactide, i.e., about 0.1% m/m) used as catalyst. The phenolic antioxidant of choice (Tyr, DBA, AA, Pyr or VA), that acted as initiator of lactide polymerization was loaded at 0.2% m/m with respect to lactide. The mixture was allowed to react under slow nitrogen flow at 180 °C for 1.5 h using mechanical stirring (50 rpm). At the end of the reaction, the polymer was left in the flask under nitrogen flow and cooled at room temperature overnight. After the syntheses, all samples underwent Solid State Polymerization (SSP) at 150 °C for 12 h under vacuum (4 mbar). All the analyses were conducted on the samples obtained after SSP. Table 1 reports all the synthesized samples, while a representative general reaction scheme is reported in Scheme 1.

### 2.3. Characterization

#### 2.3.1. Determination of Antioxidant Capacity of Polymers In Vitro

The radical scavenging capacity of the phenolic antioxidants and of the synthesized polymers was evaluated through a reaction with the stable 2,2-diphenyl-1-picrylhydrazyl radical (DPPH). Prior to analysis, all polymer samples were dissolved in DCM (0.1 g in 10 mL), precipitated in methanol and dried under vacuum (2 h, 40 °C at 2 mmHg) to eliminate traces of unreacted initiator or antioxidants blended with the polymer (in the case of the industrial sample, 4043D). For all samples, 0.05 g of polymer were cut and placed in a tube with 2 mL of 0.21 mM solution of DPPH in methanol. The tubes were vigorously vortexed for 1 min to assure full contact between the polymer and the solution. After vortexing, the tubes were left in the dark for 30 min at room temperature. The absorbance was then measured against methanol at 515 nm in 1 mL cuvettes using a spectrophotometer (UV-1700 Pharma Spec, Shimadzu, Milton Keynes, England). Due to it its spare electron, the DPPH radical shows a strong purple coloration and absorption at 515 nm. When this radical species is captured by an antioxidant, the absorption decreases and the resultant discoloration is directly proportional to the number of free radicals captured.

The radical scavenging capacity was expressed as,
%inhibition (PI) = (1− *A*_antioxidant polymer_/*A*_blanc_) × 100
where *A*_antioxidant polymer_ is the absorbance of the solution in the presence of sample, and *A*_blanc_ is the absorbance of the DPPH solution [24].

#### 2.3.2. Size Exclusion Chromatography (SEC)

SEC was performed using a system based on a Isocratic HPLC pump (Waters 1515, Waters Corporation, Milford, Massachusetts, U.S.A.) and a four column set (10^3^Å-10^4^Å-10^5^Å-500Å) (Phenomenex Phenogel, Phenomenex Inc., Torrance, California, U.S.A.) using a flow rate of 1 mL/min and 20 μL as injection volume. The detector was a Dual *λ* Absorbance Detector (Waters 2487, Waters Corporation, Milford, Massachusetts, U.S.A.), set at 230 nm. Samples were prepared dissolving 30 mg of polymer in 1 mL of anhydrous DCM; before the analysis, the solution was filtered through 0.45 μm filters. Given the relatively high loading, a check was performed using lower concentrations of polymer (5 mg/mL), in order to verify that no column overloading could be observed. Higher loadings were preferred as the UV absorption of PLA is relatively weak. *O*-dichlorobenzene was used as internal standard (peak appears at 46 min in the chromatograms). The molecular weight data were obtained using a linear polystyrene standard calibration in the range 1,600,000–106 Da. In the paper, curve height was normalized around the peak of the polymer.

#### 2.3.3. Thermal Analysis 

DSC analyses were conducted using a Mettler Toledo DSC1 (Mettler Toledo Inc., Columbus, Ohio, U.S.A.), on samples weighting from 3 to 6 mg each using a 40 µl aluminium pan. The following temperature cycles were used:Heating from 25 °C to 200 °C at 10 °C/min;5 min isotherm at 200 °C;Cooling from 200 °C to 25 °C at 10 °C/min;2 min isotherm at 25 °C;Heating from 25 °C to 200 °C at 10 °C/min;

The first two cycles were run to eliminate residual internal stresses deriving from the synthesis. Glass transition temperature (*T*_g_), cold crystallization temperature (*T*_cc_) and melting temperature (*T*_m_) were determined. The crystalline weight fraction (*χ*_c_) of the samples was determined according to Equation (1),
(1)$$\chi_{c}=\frac{∆H_{m}-∆H_{cc}}{∆H_{0}}\times 100$$
where Δ*H*_m_ is the melting enthalpy recorded on the second heating, Δ*H*_cc_ is the cold crystallization enthalpy recorded on the second heating scan and Δ*H*_0_ is the melting enthalpy of the 100% crystalline polymeric matrix (93 J g^−1^) [25].

#### 2.3.4. Thermogravimetric Analysis (TGA)

TGA were performed with a Perkin Elmer TGA 4000 (Perkin Elmer Inc., Waltham, Massachusetts, U.S.A.) under nitrogen flux at 20 mL/min with a temperature ramp from 30 °C to 550 °C at 20 °C/min on about 6 mg of samples. Isothermal analyses were performed at 80 °C for 120 min under nitrogen flux of 20 mL/min.

#### 2.3.5. Film Casting

Films were obtained via solvent-casting using the following procedure; 10 g of polymer were dissolved at room temperature in 50 g of DCM. The solution was cast on a glass surface and the solvent was evaporated overnight at room temperature and pressure. The absence of residual solvent in the films was checked via isothermal TGA (120 min at 80 °C under nitrogen flow). Film thickness was measured using a Digimatic micrometer (Mitutoyo, Sakado, Japan). The value of film thickness was obtained by averaging 10 measurements, the average value obtained being 74 ± 18µm.

#### 2.3.6. H NMR Analysis

^1^H NMR spectra were registered with a Bruker Ultrashield 400 MHz (Bruker Corporation, Billerica, Massachusetts, U.S.A.). The chemical shifts are reported in ppm and referred to TMS as internal standard. All samples were prepared by dissolving 6–8 mg of polymer in 1 mL of CDCl_3_.

### 2.4. Evaluation of Antioxidant Capacity of Films In Vivo

Antioxidant films made from PLA-VA were used for the in vivo study. The antioxidant potential of the films, in order to prevent salami oxidation during refrigerated storage, was assessed. 

#### 2.4.1. Sample Preparation

Salami (79% pork lean, 21% subcutaneous back fat, and additives, in g/kg of batter: salt, 29; sodium nitrite, 0.15; dextrose, 10; lactose, 20; black pepper, 3.0; sodium erythorbate, 0.5; carmine, 0.05; sodium caseinate, 10; soy protein, 12; decalcified water, 63, and starter culture consisting of lyophilized *Lactobacillus sakei* at levels of 10^7^ CFU/g) was sliced to 1 mm thick samples in a white chamber refrigerated at 1 °C. Samples were vacuum packed in PA/PE bags (Sacoliva, Spain) and stored at 4 ± 2 °C in a display cabinet (see Figure 1). The samples were subjected 12 h to light (fluorescent lamp) and 12 h to darkness (simulating retail conditions). Two batches were obtained: a control batch without film (C) and a batch packed with antioxidant films (8 × 8 cm^2^ PLA-VA films) used as interleaves to separate salami slices.

Three samples of each batch were removed at 0, 7, 14, 43, and 51 days of storage for analysis. The upper slice was used for color measurements, and after color measurement all the slices were minced for further analysis.

#### 2.4.2. Water Activity and pH Analysis

The pH of the minced samples was measured directly with a Crison penetration 52-32 probe connected to a Crison Basic 20 pH-meter (Crison Instruments S.A., Alella, Spain). The mean of three measurements was recorded for each sample. The water activity of the minced samples was measured using a water activity meter AquaLab™ Series 3 (Decagon Devices, Inc., Pullman, WA, USA).

#### 2.4.3. Color Measurements

Instrumental color measurement of films was performed using a Konic Chroma Meter CR-410 (Minolta, Osaka, Japan). D_65_ illuminant and 2° standard observer were chosen. *L** (lightness), *a** (redness, greenness), and *b** (yellowness, blueness) color values were determined using the 1976 CIELAB system. The chromameter was calibrated before each series of measurements using a white ceramic plate. Three different points from each sample were tested and the results averaged.

#### 2.4.4. Lipid Oxidation Analysis

Thiobarbituric acid reactive substances (TBARS) was determined following an adaptation of literature protocol [26]. Two grams of minced salami were homogenised with 20 mL of 1.2 M HCl solution containing 0.1% w/v propyl gallate and 0.1% w/v EDTA for 30 s using an ULTRA-TURRAX^®^ blender. The homogenate was centrifuged at 5,000 rpm for 10 min. The supernatant was injected in a continuous flow analyzer Futura System (Alliance Instruments, Frepillon, France). A solution of 1.2 M HCl containing 0.327% thiobarbituric acid and 0.5% Brij-35 was also injected in the system. The system consists of a bath at 90 °C were the reaction is accelerated and a colorimeter set at 531 nm to detect the reaction product malondialdehyde (MDA). The calibration curve was prepared using 1,1,3,3-tetraethoxypropane as a standard. The results were expressed as mg MDA/kg salami.

#### 2.4.5. Statistical Analysis

Statistical analysis was performed using the General Linear Model from SAS 9.2 software (Statistical Analytical Systems Institute, Cary, NC, USA). The batch (C and PLA-VA), the storage time (0, 7, 14, 43, 51 days), and their interaction were included in the model as fixed effects. Differences between effects were assessed by the Tukey test (*p* < 0.05).

## 3. Results

Given the presence of multiple –OH groups, all tested active molecules could, in principle, act as multifunctional initiators in lactide Ring Opening Polymerization (ROP). On a general level, it has been demonstrated that phenolic moieties does not interfere with initiation processes promoted by aliphatic alcohols [27]. On the other hand, the prolonged reaction times and high temperatures can favor chain transfer and backbiting reactions by the acidic phenolic –OH groups. The use of either Tyr, VA or DBA could therefore end up in the formation of products with similar molecular weights, given the presence of just one aliphatic alcoholic moiety in each molecule.

### 3.1. SEC Results

Data obtained via SEC (${\bar{M}}_{n}$, ${\bar{M}}_{w}$ and dispersity *Đ*) on the polymers synthesized are given in Table 2. Natureworks^®^ Ingeo 4043D was used as an industrial reference standard. The table also reports the coloration of the samples obtained. Figure 2 shows the SEC curves of 4043D and PLA-VA, i.e., the best polymer, in terms of molecular weight, molecular weight distribution and virtual absence of low molecular weight species: no residual unreacted vanillyl alcohol was detected. The curves of other samples are reported in Supporting Information (Appendix A).

The reported differences in terms of molecular weights and dispersity can be ascribed to the presence of multiple phenolic –OH groups in DBA and –OMe in VA, which can influence the reaction outcome through the aforementioned side reactions. PLA-Pyr sample showed a significant decrease in molecular weight, compared to other samples, probably due to a significant degradation of Pyr in the reaction conditions, as evidenced by the coloration of the polymer obtained. PLA-AA product was characterized by significantly higher molecular weight, coupled with high polydispersity. This result was attributed to the occurrence of side reactions resulting in branching and, ultimately, crosslinking likely due to the AA double bond. This hypothesis was confirmed by the presence of an insoluble fraction retained on the filter used before the injection.

### 3.2. DPPH Test Results

As indicated in Table 2 PLA-AA, PLA-DBA and PLA-Pyr above all, showed coloration of the polymer after the synthesis, indicating at least partial degradation of the polymer itself. DPPH tests were, therefore, conducted on PLA-VA and PLA-Tyr to assess their potential antioxidant feature. Results are shown in Table 3.

PLA-VA results the best one in terms of radical scavenging on DPPH and for this reason it was chosen as the polymer to be used for the in vivo experiments. Thermal analyses were conducted on PLA-VA and compared to 4043D to assess the thermal features of the sample in comparison to an industrial PLA.

### 3.3. H NMR Analysis

In order to fully assess the activity of VA as an initiator in the ring opening polymerization of lactide, a reaction with 4% m/m of VA, with respect to lactide, was carried out using the same conditions described in paragraph 2.2. The product purified through dissolution in methylene chloride and precipitation in methanol, in order to remove potentially unreacted vanillyl alcohol. The purified product was analysed through ^1^H NMR. High loading of VA yielded a low molecular weight product, but allowed a more precise structural investigation, given the high concentration of initiator and of chain-end groups. ^1^H NMR spectrum of the product was reported in the SI file (Appendix A), with magnification of the area of interest (Appendix A). Two main peaks were detectable, namely a quartet centered at 5.17 ppm, given by the methine group of the lactide-derived repeating units and a doublet centered at 1.60 ppm, relative to the corresponding methyl group. As Appendix A shows, signals were detected in the 7.20–6.60 region, accounting for the aromatic protons of VA. The quartet centered at 4.37 ppm was attributed to the methine group of the chain-end, while the broad signal at 3.82 ppm to the methoxy group on the VA ring. The absence of signals related to the –CH_2_–OH group, that should fall at about 4.3 ppm [28] confirms that the initiator reacted only with this alcohol and not via the phenolic moiety. The consistency of the integrals relative to these two signals likely demonstrates that all the chains were initiated by a vanillyl alcohol unit. In addition, the concentration of chain-end groups was calculated as the ratio between the area of the quartet at 4.37 ppm and the sum of the integrals of the signals at 5.17 and 4.37 ppm, pointing to a concentration of 2% m/m of chain-end groups versus the lactic acid repeating unit and demonstrating a quantitative reaction of VA, as initiating a species in the ROP of lactide. No signals related to unreacted vanillyl alcohol are visible.

### 3.4. Thermal Analyses

#### 3.4.1. Differential Scanning Calorimetry (DSC)

Glass transition temperature (*T*_g_), cold crystallization temperature (*T*_cc_), cold crystallization enthalpy (Δ*H*_cc_), melting temperature (*T*_m_), melting enthalpy (Δ*H*_m_) and crystallinity (*χ*_c_) data of PLA-VA and 4043D are provided in Table 4; all the data reported referred to the second heating scan. Thermograms are reported in Supporting Information (Appendix A).

Both samples show a clear *T*_g_, in the same temperature range. PLA-VA thermogram is characterized by an exothermic peak related to the cold crystallization and a melting peak. Both samples are highly amorphous, with PLA-VA being slightly more crystalline than 4043D. In particular, the latter shows a visible melting peak in the first heating scan that is barely visible during the second heating scan, indicating that 4043D le cannot efficiently crystallize when cooled from the melt. It is worth noting that, even if *χ*_c_ is very low for PLA-VA, as common in PLA-based polymers [29], and shows high cold crystallization and melting enthalpies, indicating a higher tendency to crystallize in comparison to 4043D. The lower crystallinity of PLA Ingeo 4043D is likely due to the higher d-lactide stereisomer content (less than 1% in l-lactide used for the synthesis of PLA-VA, while producer declares a 4.3% of d-lactide in 4043D). 

#### 3.4.2. Thermogravimetric Analysis (TGA)

Thermogravimetric analysis (TGA) data are reported in Table 5.

Thermal stability of PLA-VA was also assessed in comparison to 4043D checking the temperature of 1%, 5%, 50% and 95% of weight loss (*T*_1%_, *T*_5%_, *T*_50%_, and *T*_95%_ respectively). 4043D starts to degrade at approximately 320 °C, almost 50 °C higher than PLA-VA. Although, the data show that PLA-VA have a lower thermal stability compared with 4043D, melt processing should not be affected, as degradation starts at higher temperatures than those used for standard PLA processing, (190–200 °C for melt extrusion or injection molding). TGA degradation profiles are reported in Figure 3.

### 3.5. Validation of Antioxidant Capacity of Films in vivo

#### 3.5.1. Water Activity and pH

The water activity (*a*_w_) and pH values of salami at the beginning of the study (*t*_0_) were 0.911 ± 0.003 and 4.7 ± 0.05, respectively. No significant changes on the *a*_w_ and pH values were observed throughout 51 days of refrigerated storage of salami (data not shown). Packaging of salami slices using PLA-VA films as interleaves did not alter (*p* > 0.05) the *a*_w_ and pH values of salami slices (C) during storage. 

#### 3.5.2. Color Measurements

The evolution of color parameters (*L**, *a**, and *b** values) of salami samples throughout refrigerated storage under light exposure are given in Table 6. The lightness (*L**) of control samples decreased (*p* < 0.05) from day 43 of storage in refrigerated display cabinets, while salami packed with PLA-VA films showed no significant changes on *L** values throughout its shelf life. Redness (*a**) and yellowness (*b**) coordinates of salami significantly decreased during storage of both, C and PLA-VA samples. However, salami samples packed with PLA-VA films showed higher *a** and *b** values (*p* < 0.05) than control samples from day 43 until the end of refrigerated storage. The formation of the typical cured color is due to nitrosylmyoglobin, a pigment resulting of the reaction of nitrite with meat myoglobin. Meat discoloration (loss of red color intensity) is mainly due to the reaction of nitrosylmyoglobin with the residual oxygen present in the package [30]. Light exposure accelerates oxidation reactions of haem pigments, and therefore, meat discoloration. On the other hand, changes in *b** coordinate in dry fermented sausages are usually related to lipid oxidation [31].

The reported results would suggest that packaging with PLA-VA films would reduce pigment and lipid oxidation, and would contribute to improve color stability of salami during refrigerated storage under light exposure (Figure 3).

#### 3.5.3. Lipid Oxidation

The extent of lipid oxidation undergone during the shelf life of salami was measured as thiobarbituric acid reactive substances (TBARS). TBARS values of all salami samples significantly increased throughout refrigerated storage under commercial illumination conditions (Figure 4). Control samples showed a more pronounced increase of lipid oxidation throughout the study, resulting in higher TBARS values than PLA-VA samples (*p* < 0.05) for all studied sampling times. The samples packed with PLA-VA films showed no increase in TBARS values during the first 14 days of storage, thereby contributing to an extended shelf life for this type of product.

Several authors have reported the benefits of packaging, with added antioxidants, to improve color and oxidative stability of meat and meat products [32,33,34]. However, to the best of our knowledge, no antioxidant food packaging systems, with crafted antioxidants, have been tested. 

## 4. Conclusions

The use of biobased antioxidant compounds as initiators in the ROP of lactide, yielding PLA-based materials, carrying covalently bound antioxidant moieties was successfully demonstrated. The products were analysed in terms of both, molecular weights and thermal properties, and compared with a commercial grade PLA used for the production of flexible films (Natureworks 4043D). Among the synthesized samples, PLA initiated with vanillyl alcohol (PLA-VA) displayed antioxidant features when tested in vitro through DPPH protocol. In addition, PLA-VA molecular weight and thermal properties were found to be comparable to the ones of 4043D, further demonstrating the industrial viability of the synthetic protocol here presented. Finally, PLA-VA films obtained via solution casting were tested also through in vivo experiments using it as packaging for salami, demonstrating its effectiveness in improving color stability and reducing lipid oxidation of salami during refrigerated storage. This results might pave the way for the use of PLA-VA as a promising intrinsically antioxidant material for flexible food packaging applications.

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
