# Peer review of "Synthesis of Polylactic Acid Initiated through Biobased Antioxidants: Towards Intrinsically Active Food Packaging"

_polymers, 2020, doi:10.3390/polym12051183_

Round 1
Reviewer 1 Report
This is an interesting paper reporting the synthesize of poly(L-lactide) containing antioxidant covalently linked with a polymer chain via ROP of L-Lactide using bio-based phenolic-based antioxidants as initiators in conjunction with tin octoate as catalyst. The antioxidant properties of the synthesized polylactides, especially those prepared using vanillyl alcohol as an initiator (PLA-VA), were confirmed by in vitro tests. Additionally, the efficiency of polylactide films obtained from PLA-VA in improving the color stability and reducing of lipid oxidation was confirmed by in vivo experiments using the film as packaging for salami. Based on the above mentioned, it may be conclude that PLA-VA is a promising material possessing antioxidant properties for packaging applications. However, this paper suffers from insufficient confirmation of incorporation of antioxidants into polymer chain. 1H NMR spectra should be presented to confirm that bio-based substituted phenols are indeed incorporated into polymer chain. A separate set of experiments with lower monomer to initiator ratio could be performed to synthesize PLA with lower molecular weight that will facilitate the NMR and, possible, MALDI TOF MS, analyzes of synthesized polymers. In addition, the effect of the phenol derivatives on the molecular weight and polydispersity of synthesized polymers is required deeper investigation. Evidently, in VA the methoxy group in para-position to hydroxyl group reduces the acidity of VA in comparison with phenol or phenol derivatives studied here. High polydispersity of PLA prepared with AA as initiator could be explained by the presence of primary and secondary hydroxyl groups of different reactivity in ROP of lactide. Probably, the investigation of interaction of phenols with catalyst (Sn(Oct)2) will give useful information about side reactions occurring during the polymerization in the presence some of these phenols. Therefore, I recommended major revision for this paper.
Author Response
We thank the reviewer for the very valuable and interesting comments. In order to answer to the need of deeper structural investigation, we performed an additional reaction with higher vanillyl alcohol (VA) loading (a twenty-fold increase in concentration with respect to previous reactions, i.e. 4% m/m on lactide). The 1H NMR analysis of the purified product was useful to demonstrate the presence of VA in the final polymer, after precipitation. Given the high loading of initiator, the overall molecular weight of the product was low, allowing to properly identify the chain-end groups. This investigation would’ve been impossible on the samples bearing the 0.2% of initiator. It was possible to demonstrate that the concentration of lactide-derived chain-ends was equal to the concentration of VA in the product and calculated to be of 4% m/m. This indicates the quantitative reaction of the VA in the tested conditions. As the reviewer pointed out, the polymerization reaction of lactide catalyzed by Sn(Oct)2 is very sensitive to the presence of acidic species, which are known to deactivate the catalyst. Given the results obtained at high VA concentration, we believe that the phenol proton acidity doesn’t affect significantly the reactivity of the system.
We added a paragraph related to NMR study in the paper and the NMR spectrum in the SI file
Reviewer 2 Report
Comments on: “Polymers”
“Synthesis of Polylactic acid initiated via biobased antioxidants: towards intrinsically active food packaging”
Marco Aldo Ortenzi, Stefano Gazzotti, Begonya Marcos Muntal, Stefano Antenucci, Stefano Camazzola, Luciano Piergiovanni, Hermes Farina, Giuseppe Di Silvestro and Luisella Verotta
The authors describe the use of bio-based phenols as initiators in the ROP of lactide, affording PLAs with chemically bound antioxidant moieties. These materials were tested both “in vitro” and “in vivo” in order to assess their antioxidant features. In this regard, promising results have been obtained in the case of the polymer prepared with vanillyl alcohol as initiator. Indeed, it exhibited better antioxidant behavior compared to a commercial sample with similar molecular weight and thermal properties.
Given the novelty of the research and the remarkable results achieved, I recommend the publication of manuscript in “Polymers”. However, the following point need to be addressed.
- The 1H NMR spectra, at least for the PLA-VA and the commercial sample, should be reported. I think the resonances corresponding to the VA moiety are easily identifiable and the signal integrations could also provide an estimation of the antioxidant-group content.
- Figures 1 and 2 should be readable also if printed in black and white. Please consider the use of dashed lines/markers to identify the different curves.
- Concerning the Mn values reported in Table 2, I wonder if Mark-Houwink correction factors applicable to the analytical conditions employed (CH2Cl2, PS standards) are available.
Minor comments:
Page 7, line 245: “Compared to” would read better than “with respect to”
Page 11, line 358: Please remove the sentence “For research articles with several authors, a short paragraph specifying their individual 359 contributions must be provided. The following statements should be used”
Author Response
The authors describe the use of bio-based phenols as initiators in the ROP of lactide, affording PLAs with chemically bound antioxidant moieties. These materials were tested both “in vitro” and “in vivo” in order to assess their antioxidant features. In this regard, promising results have been obtained in the case of the polymer prepared with vanillyl alcohol as initiator. Indeed, it exhibited better antioxidant behavior compared to a commercial sample with similar molecular weight and thermal properties.
We wish to thank the reviewer for the very positive comments: corrections made according to his/her observations are given here below
Given the novelty of the research and the remarkable results achieved, I recommend the publication of manuscript in “Polymers”. However, the following point need to be addressed.
- The 1H NMR spectra, at least for the PLA-VA and the commercial sample, should be reported. I think the resonances corresponding to the VA moiety are easily identifiable and the signal integrations could also provide an estimation of the antioxidant-group content.
Given the low amount of active principle loaded, we believed that a reliable quantification of the bounded molecule would’ve been difficult to accomplish. For this reason, we performed an additional reaction with higher VA loading, in order to demonstrate the reactivity of VA itself as initiator. The 1H NMR spectrum of the product was reported, with assignation of the peaks and results were added and discussed in the paper.
- Figures 1 and 2 should be readable also if printed in black and white. Please consider the use of dashed lines/markers to identify the different curves.
According to the suggestion, we modified the two figures as you can see in Figures 2 and 3.
- Concerning the Mn values reported in Table 2, I wonder if Mark-Houwink correction factors applicable to the analytical conditions employed (CH2Cl2, PS standards) are available.
We thank the reviewer for this comment: we also tried to search for this kind of correlation during the writing of the paper but, unfortunately we could only find Mark-Houwink correction factors in THF for L-LA, that is 0.58 according to Slattery, R.M. et al. Journal of Polymer Science, Part A: Polymer Chemistry, (2019), 57, 48-59 DOI: 10.1002/pola29280 citing as source for the value a research from Prof. Dubois group, i.e. I. Barakat, P. Dubois, R. Jerome, P. Teyssie, J. Polym. Sci. Part A: Polym. Chem. (1993), 31, 505-514, DOI: 10.1002/pola.1993.080310222.
Also Al-Khafaji, Y.F. et al report the same value for THF in RSC Advances, (2017), 7, 4510-4517 DOI: 10.1039/c6ra26746g but use it for synthesis performed using racemic lactide.
It has to be considered that according to a well-known review by Garlotta, D. Journal of Polymers and the Environment, (2001), 9, 63-84 DOI: 10.1023/A:1020200822435 Mark-Houwink constants change according not only to the solvent used but also to the stereochemistry of the polymer (Table XVIII in the review). Therefore we expect that, in our case, corrections should be slightly different for 4043D (that has 4.3% of D isomer) and for our samples, that derive from L-lactide.
Backes, E.H. et al Journal of Composites Science, (2019), 3, 52 DOI: 10.3390/jcs3020052 declare for 4043D an Mn of 106000 Da in THF (PS calibration) that lowers to 74000 Da after processing but no indication about columns set (number of columns and type) is given. Since Dispersity results relatively low (around 1.6) we do not consider the values indicated as comparable to ours.
Minor comments:
Page 7, line 245: “Compared to” would read better than “with respect to”
Done
Page 11, line 358: Please remove the sentence “For research articles with several authors, a short paragraph specifying their individual 359 contributions must be provided. The following statements should be used”
Done
Reviewer 3 Report
Comments for authors on manuscript polymers-785114: "Synthesis of Polylactic acid initiated via biobased antioxidants: towards intrinsically active food 3 packaging” by M. Aldo Ortenzi, S. Gazzotti, B. Marcos Muntal, S. Antenucci, S. Camazzola, L. Piergiovanni, H. Farina, G. Di Silvestro, and L. Verotta.
In this manuscript the authors describe the synthesis and characterization of new biodegradable polymers functionalized with biobased antioxidants by bulk ring-opening polymerization of L-LA, using tin octanoate as catalyst with different phenolic antioxidants as initiators. These new polymers are used as potentially biodegradable material for food packaging applications. The paper is well arranged for me. The paper is suitable for publication in Polymers after minor revision:
-The quality and clarity of presentation should be improved: the units to express the amount of catalyst or initiator must be equal (on page 3, Synthesis of phenol-functionalised PLA samples: “…were added in the feed together with tin octanoate (0.3% w/w on lactide) used as 100 catalyst. The phenolic antioxidant of choice (Tyr, DBA, AA, Pyr or VA), that acted as initiator of 101 lactide polymerization was loaded at 0.2% m/m with respect to lactide…”; The The numbering of the tables is not uniform: see for example Table 2 on page 6 and Table II on page 7, or Table V on page 8 or Table VI on page 10.
- The use of this polymers as biodegradable material for food packaging application have the problem that the authors have used a tin based-catalyts. As it is known, tin compounds have high toxicity. There are a lot of report catalyst from biocompatible and non-toxic metals compounds such as Zn or Mg compound which can be use as catalyst for the ROP of L-LA, rac-LA…The authors will be able to use this type of less toxic catalysts than those of tin.
- The author should attempt to characterize the polymers by NMR.
Author Response
In this manuscript the authors describe the synthesis and characterization of new biodegradable polymers functionalized with biobased antioxidants by bulk ring-opening polymerization of L-LA, using tin octanoate as catalyst with different phenolic antioxidants as initiators. These new polymers are used as potentially biodegradable material for food packaging applications. The paper is well arranged for me. The paper is suitable for publication in Polymers after minor revision:
We wish to thank the reviewer for the very positive comments regarding the paper.
-The quality and clarity of presentation should be improved: the units to express the amount of catalyst or initiator must be equal (on page 3, Synthesis of phenol-functionalised PLA samples: “…were added in the feed together with tin octanoate (0.3% w/w on lactide) used as 100 catalyst. The phenolic antioxidant of choice (Tyr, DBA, AA, Pyr or VA), that acted as initiator of 101 lactide polymerization was loaded at 0.2% m/m with respect to lactide…”; The The numbering of the tables is not uniform: see for example Table 2 on page 6 and Table II on page 7, or Table V on page 8 or Table VI on page 10.
The reviewer is right: we corrected, deciding to keep roman numbers for tables (I, II, III etc).
Regarding catalyst, usually for PLA it is indicated in terms of w/w but it is true that this is not in line with the quantities indicated for each initiator therefore we added the m/m of catalyst vs. lactide.
- The use of this polymers as biodegradable material for food packaging application have the problem that the authors have used a tin based-catalyts. As it is known, tin compounds have high toxicity. There are a lot of report catalyst from biocompatible and non-toxic metals compounds such as Zn or Mg compound which can be use as catalyst for the ROP of L-LA, rac-LA…The authors will be able to use this type of less toxic catalysts than those of tin.
The reviewer is right, tin is toxic: anyway, stannous octanoate is the catalyst most frequently used for the industrial synthesis of PLA because it is soluble in molten lactide and gives high conversion and very low racemization. It is thus preferred when a PLA with some crystallinity is necessary (ex. for good mechanical properties). In past experience we used other catalysts that did not give the results we obtained with stannous octanoate.
Usually, due to its toxicity, when using the industrial process the catalyst is “killed” (i.e. it is deactivated) using the so-called “Cat-killer” molecules, usually phosphoric acid or acetic anhydride and then it is removed almost completely using a stripping phase. The molten PLA is put under vacuum with a flux of nitrogen and this allows removing the catalyst (that is combined with the inhibitor) and the residual lactide and quantities of Sn <10 ppm and of lactide <0.3% w/w can be achieved. A good explanation of the issues related to tin and of the ways it can be removed from PLA is present in “Polylactic Acid: A Practical Guide for the Processing, Manufacturing, and Applications of PLA” by Lee Tin Sin and Bee Soo Tueen, edited by Elsevier. We used the Second Edition and some good description of the processes related to stannous octanoate are present in pages 73-75
Since we cannot perform such steps on a lab scale, we used the PLA we obtained without removing stannous octanoate. In the future we would like to keep on testing also other catalysts in order to reduce the problems due to tin, using organic molecules.
- The author should attempt to characterize the polymers by NMR.
According to the request, we added the 1H NMR spectrum of PLA-VA sample. In particular, we performed a synthesis with higher loading of VA, in order to perform a proper and reliable structural investigation and to calculate the actual quantity of bounded active principle after purification.
Reviewer 4 Report
Dear Authors,
Please find my comments as attached.

Author Response
In the submitted paper, new Polylactide (PLA)-based polymers have been synthesized, and functionalized with biobased antioxidants, to reach active, biobased and potentially biodegradable material for food packaging applications. The materials varied in phenolic antioxidants as initiators in the ring opening polymerization of L-lactide. The main emphasis was first given to recognize the most promising synthesized polymer among all the variations through different analytic methods while these polymers were also compared with the ones of a PLA Natureworks 4043D. After choosing PLA-VA as the most promising polymer bearing vanillyl alcohol as initiator, this polymer was then evaluated further by different analytical methods for active food packaging applications. The paper provides a detailed chemical analysis of the new bio-based materials’ structure, and some new data concerning their nature. Its originality and quality are high enough to recommend the paper for publication. However, some minor revisions are suggested. In following please find my specific comments:
We wish to thank the reviewer for the very positive comments.
- The authors have suggested on page 7 that high Mw and high PI are related to the crosslinking and/or branching of the double bonds present in the molecule. I’m wondering if any detailed studies by 1H NMR and 13C NMR were done to identify and confirm the chemical structure / microstructure of the synthesized polymers as well as to confirm the hypothesis of the side reactions’ occurrence?
We thank the reviewer for this observation, that is very useful.
We made some studies via NMR on PLA containing Vanillyl alcohol since it was the polymer of choice for the in vivo tests. Since the polymers we synthesized have very low quantities of initiator (0.2% m/m on lactide, i.e. 0.1 m/m on the lactic acid repeating unit) that do not make it easy to perform a valid NMR study, we performed H-NMR on a PLA-VA containing 2% m/m synthesized on purpose, to confirm that VA fully reacts and is not lost during the polymerization: now this information has been added in the paper and the NMR is reported in SI. The other samples described in the paper, namely PLA-Pyr, PLA-Tyr, PLA-DBA and PLA-AA were not analyzed via NMR. PLA-AA is coloured and is not soluble in methylene chloride and chloroform, therefore it could not be analyzed via NMR. This made us think that crosslinking occurs, probably due to the double bonds present. PLA-Pyr is heavily coloured and with low MW therefore we concluded Pyridoxine degradates during the synthesis.
- I’m also wondering why the FTIR analysis was not employed to identify the characteristic spectra of such polymers to extend the structural characteristic of the new materials? Would be good to do some FTIR analysis as well.
We made FT-IR of the samples but, due to the very low concentration of the initiators, we could not detect any signal besides the ones of PLA. Anyway we added FT-IR of 4043D and of PLA-VA in the SI file.
- Why Salami is the only food to have been tested for these experiments? Would have been better to cover food varieties to test the characteristic of such polymers and relaying not only on the results derived from Salami for example.
This is a very good point. We tested only salami for two reasons:
- We obtained PLA films via casting from solution, letting the solvent evaporate overnight. To have enough films for testing salami, we had to prepare 120 films (two per night, having two different working position to prepare them) and it took more than two months. We wanted to test also other kind of food but it would take too much time. Now we are trying to improve the timing for preparation, in order to be able to make the production of films faster.
We have a small twin screw extruder (screws diameter = 11 mm) but unfortunately with 50 grams of polymer it is not possible to use melt extrusion to produce films suitable for this kind of tests.
2) For our experience, for this type of application it is necessary tight direct contact between the active film and the product, also high ratio of area of contact vs food weight is necessary. The food packaging system that better fits with this features is vacuum packaging of sliced food products. The election of salami is because it usually presents quality defects during refrigerated storage under lighting conditions due to fat and color oxidation.
Due to the good results obtained we are planning tests with also other foods
- Please check your text carefully, there were small mistakes here and there in the text.
Thank you. We revised the manuscript and corrected some small mistakes. We hope that now the text is better
Round 2
Reviewer 1 Report
I satisfied with the corrections made by authors. The manuscript can be accepted for publication now.